# Carbon Nanotube Modified by (O, N, P) Atoms as Effective Catalysts for Electroreduction of Oxygen in Alkaline Media

**Vera Bogdanovskaya** [1],*, **Inna Vernigor** [1], **Marina Radina** [1], **Vladimir Andreev** [1], **Oleg Korchagin** [1] **and Vasilii Novikov** [2]

[1]   A.N. Frumkin Institute of Physical Chemistry and Electrochemistry, Russian Academy of Sciences, 119071 Moscow, Russia; msnoviinna@gmail.com (I.V.); merenkovamarina@mail.ru (M.R.); vandr@phyche.ac.ru (V.A.); oleg-kor83@mail.ru (O.K.)

[2]   D.I. Mendeleev Russian University of Chemical Technology, Miusskaya pl. 9, 125047 Moscow, Russia; nvt46@yandex.ru

*   Correspondence: bogd@elchem.ac.ru

**Abstract:** The influence of the types and amounts of oxygen (O), nitrogen (N), and/or phosphorus (P) heteroatoms on the surface of carbon nanotubes (CNTs) on stability and catalytic activity in the oxygen reduction reaction (ORR) was investigated in alkaline media. It is shown that functionalization of CNTs leads to growth of the electrochemically active surface and to an increase in activity in the ORR. At the same time, a decrease in stability is observed after functionalization of CNTs under accelerated corrosion testing in alkaline media. These results are most significant on CNTs after functionalization in $HNO_3$, due to the formation of a large number of structural defects. However, subsequent doping with N and/or P atoms provides a further activity increase and enhances the corrosion stability of CNTs. Thus, as shown by the studies of characteristic parameters (electrochemical active surface values ($S_{EAS}$); $E_{1/2}$; corrosion stability), CNTs doped with N and NP are promising catalytic systems that can be recommended for use as fuel cell cathodes. An important condition for effective doping is the synthesis of carboxyl and carbonyl oxygen-containing groups on the surface of CNTs.

**Keywords:** carbon nanotube; functionalization; heteroatoms; electrochemically active surface; oxygen reduction reaction; corrosion stability; alkaline media

## 1. Introduction

Oxygen electroreduction is one of the most important reactions due to widespread demand for its practical use. The oxygen reduction reaction (ORR) occurs with a significant overvoltage, and the reaction path depends on the nature of the electrode material. The development of active and stable catalysts for the cathodic oxygen reduction reaction is a priority problem for the production of fuel cells (FC) and metal-air batteries [1,2]. Among such systems, alkaline FCs operating with high efficiency at atmospheric pressure in a wide temperature range are of particular importance [3]. Therefore, studies were carried out in an alkaline electrolyte.

A number of studies have shown that carbon nanotubes (CNTs) are promising materials for FC cathodes due to such properties as corrosion stability, large specific surface area, and high electric conductivity [1,4]. However, initial CNTs have low activity in ORRs, and it is therefore necessary to modify their surfaces in order to create active centers (AC) [5]. The most common approaches for FC development are functionalization (oxidation) of the CNT surface using acids or bases, and doping with heteroatoms. Functionalization promotes formation of functional oxygen-containing groups on the surface of CNTs that provide an increase in activity in ORRs [6–8]. The introduction of

heteroatoms, such as nitrogen, phosphorus, sulfur, etc., into the structures of CNTs leads to a further activity increase as a result of a change in the electronic structure of carbon material [9,10]. For example, during doping with nitrogen that acts as an electron donor in carbon material, the Fermi level shifts into the conduction band, which makes all of the nitrogen-doped carbon material metallic. This facilitates the transfer of electrons to adsorbed oxygen molecules, which is consistent with the density functional theory [11]. The difference in electronegativity, regardless of whether the dopants have a higher (as N) or lower (as B, P, S) electronegativity than that of carbon, causes a change in the charge distribution favoring $O_2$ adsorption [11,12]. This promotes the breaking of O-O bonds in oxygen molecules and increases the selectivity of the reaction of oxygen reduction to water. A number of studies [13,14] reported that carbon doped with both N and P atoms exhibits higher electrocatalytic activity in ORRs and demonstrates improved oxygen adsorption, which, according to the authors, may be due to a synergistic interaction between N and P, or just to the effect of the combination of single N and P atoms. In [13], doping of CNTs with nitrogen and phosphorus atoms was carried out directly during synthesis of CNTs (Ar, 760–840 °C) in the presence of benzylamine and triphenylphosphine (TPP) as sources of nitrogen and phosphorus, respectively, and ferrocene as a catalyst. In [14], a post-treatment method was adopted to prepare dual-doped carbon by heat treatment (800 °C) of mesoporous carbon in an inert zone, where TPP and dicyandiamide (DCDA) were used as P and N precursors. The total amount (at. %) of N and P on the surface of the carbon material under these conditions ranged from 4.08 (simultaneous doping) and 4.1 (first N, then P) to 3.74 (first doped with P, then N). Here, an increase in activity in ORRs was observed with the introduction of two atoms, especially in the latter case, with an onset potential of 0.88 V. The authors attribute this to the fact that in the presence of P, N atoms are introduced in the graphitic N positions that are most active in ORRs. This type of nitrogen is possibly formed during partial replacement of phosphorus. However, these assumptions require additional confirmation, and the presence of oxygen-containing groups on the surface should also be taken into account.

Many articles [6–10,14] devoted to the electrocatalysis of ORRs as a result of research present cyclic voltammogram (CV) and polarization curves, on the basis of which they make conclusions about an increase in activity. However, a sufficient number of parameters calculated on the basis of CVs and polarization curves are not given. It is difficult to compare electrochemical characteristics from different papers because results were obtained under different experimental conditions.

Much less attention is paid to the corrosion stability of CNTs and the influence of the amount and method of doping on stability, as well as the necessity of pre-functionalization [15]. Such studies are required because structure defects can be formed in the processes of functionalization and doping, which cause decreases in the BET (Brunauer–Emmett–Teller) surface area, pore size, and surface area.

The aim of this study was to determine the effects of modification of CNTs with oxygen, nitrogen, and/or phosphorus atoms on stability and activity in the ORR in alkaline electrolytes, and to synthesize a catalyst free of precious metals for use as a fuel cell cathode.

## 2. Results and Discussion

The results of structural and electrochemical studies of CNTs subjected to functionalization and subsequent doping are presented.

### 2.1. X-ray Photoelectron Spectrum (XPS) Studies

In Figure 1a, the effect on the structure of the initial CNT was insignificant after treatment in 1 M NaOH. Only one type of oxygen-containing group, i.e., the hydroxyl group, was present on the surface (binding energy = 533.34 eV). When functionalization was carried out in concentrated $HNO_3$ at 120 °C for 1 h, strong surface oxidation occurred, and several types of oxygen-containing groups formed (Figure 1b). Upon separation into individual components, 1s XPS spectra of the components corresponded to different binding energies: 531.87 (ketone, carbonyl), 533.3 (hydroxyl), and 534.89 eV (oxygen-containing carboxyl groups) [16,17]. Comparison of obtained data with the literature [18]

showed that upon doping, the number of oxygen-containing groups decreased, and heteroatoms were incorporated into the carbon structure. The electron binding energy showed that N was present in the form of pyrrole and pyridine and was bonded to two C atoms, based on a comparison of electron binding energies. Binding energies of phosphorous are 133.6 and 134.4 eV, which agrees with [18].

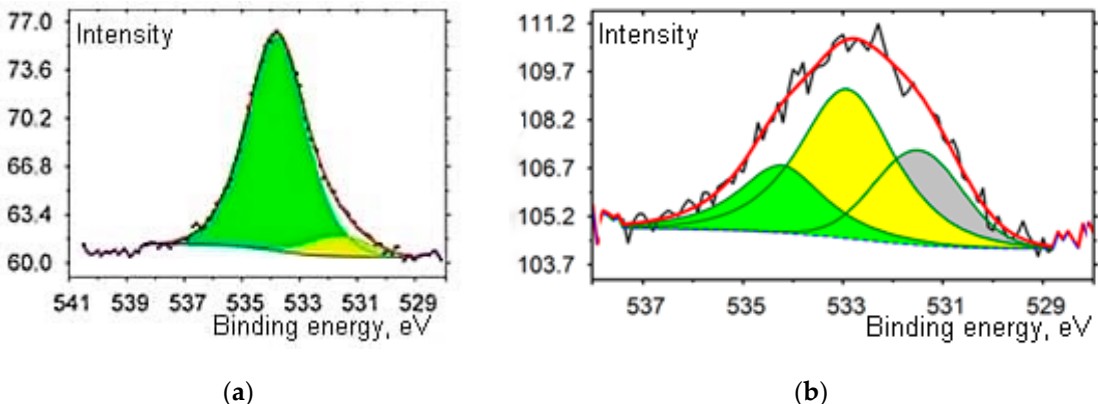

**Figure 1.** O1s X-ray spectra of carbon nanotubes (CNTs) functionalized with oxygen-containing groups after treatment in NaOH– (**a**) and in $HNO_3$– (**b**).

## 2.2. Electrochemical Studies

Electrochemical active surface values ($S_{EAS}$) are shown in Table 1. The highest value of $Q$ was achieved after functionalization with acid (CNT2). In this case, $S_{EAS}$ increased by 3.5 times in comparison with the initial CNT. This is explained by the highest overall oxygen content on the surface (Table 1) of the modified CNTs. Furthermore, characteristic peaks at the potential of 0.6 V were observed in the voltammetric curves of CNT2 (Figure 2b), which correspond to quinone-hydroquinone conversion [19]. In the XPS spectrum of CNT2, there were binding energy maxima between 531.5 and 531.9 eV, which can correspond to the C=O groups responsible for this conversion [16,20]. For CNT1 treated with alkali, $S_{EAS}$ increased only by 1.5 times in comparison with the initial CNT, and the overall oxygen content on its surface was 2.18 at. % (Table 1).

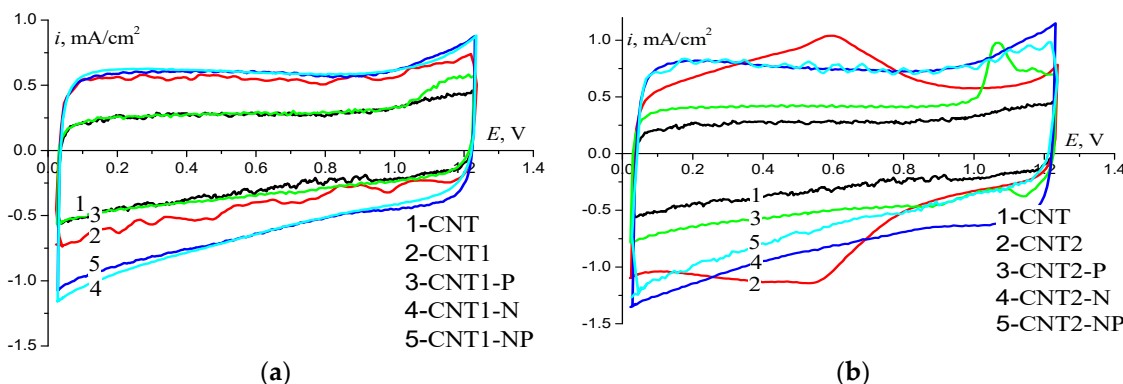

**Figure 2.** Cyclic voltammogram (CV) on CNTs doped with N and P. (**a**) –CNT functionalized with NaOH; (**b**) –CNT functionalized with $HNO_3$. Ar-saturated 0.1 M KOH, the potential scan rate is 100 mV/s; $m_{cnt}$ = 0.15 mg/cm$^2$.

**Table 1.** Characteristic parameters of CNTs after various types of modification.

| Nos. | Material | $Q$, C/g (as $S_{EAS}$) | $S_{BET}$, m²/g | $E_{1/2}$, V | $i_{kin}$, A/cm² (E, V) * | $i_{lim}$, A/cm² at 0.4 V (n) ** | Element/Overall Content, at. % |
|---|---|---|---|---|---|---|---|
| | | | | | 157 rad/s | | |
| 1 | CNT | 23 | 320 | 0.64 | 0.1 (0.750) | 2 (1.4) | O/0.46 |
| 2 | CNT 1 | 37.5 | 297 | 0.66 | 0.13 (0.750) | 2.2 (1.7) | O/2.18 |
| 3 | CNT 1–P | 25 | - | 0.72 | 0.6 (0.777) | 2.5 (1.8) | O/11.1 N/0.70 P/0.40 |
| 4 | CNT 1–N | 47.5 | 269 | 0.79 | 1.02 (0.813) | 3.62 (2.6) | O/10.08 N/1.15 |
| 5 | CNT 1–NP | 48 | 172.2 | 0.79 | 1.02 (0.819) | 3.36 (2.5) | O/2.59 N/1.49 P/0.82 |
| 6 | CNT 2 | 78 | - | 0.71 | 0.23 (0.800) | 2.42 (1.8) | O/15.4 N/1.2 |
| 7 | CNT 2–P | 37.5 | - | 0.74 | 0.49 (0.805) | 2.9 (2) | O/10.8 N/1.0 P/0.2 |
| 8 | CNT 2–N | 61 | - | 0.79 | 0.92 (0.815) | 4.2 (3) | O/12.84 N/1.98 |
| 9 | CNT 2–NP | 54 | 216.6 | 0.78 | 1.19 (0.823) | 3.75 (2.7) | O/10.8 N/1.55 P/0.4 |

* Potential values for the polarization of 0.05 V from the steady-state potential established on modified CNTs in an O₂-saturated electrolyte corresponding to $i_{kin}$ values; ** n-electron transfer number.

Further introduction of P into the structure of CNT resulted in a decrease in $Q$ in both cases of pre-functionalization (Table 1), and a peak in the anodic region at the potential of E = 1.1 V was observed in the CV (Figure 2) that was most pronounced on CNT2–P. Of all the doped CNT, CNT–N had the highest $S_{EAS}$ value, which shows that nitrogen-containing functional groups make a greater contribution to the value of the electrochemically active surface than phosphorus-containing groups. $Q$ values for dual-doped CNTs are comparable with the $Q$ values for CNT-N. However, the $Q$ values for CNT–NP and CNT–N for doped CNT2 were larger than that for doped CNT1. The shapes of CVs for these materials in the two cases of pre-functionalization are similar.

In addition, it should be noted that there was no direct correlation between the content of oxygen-containing groups on the CNT surface and $S_{EAS}$ value (Table 1). Thus, when the oxygen content on CNT1 was 2.18 at. %, the $S_{EAS}$ value was 75 C/g, while on CNT2 with an oxygen content of 15.4 at. % (seven times that compared to CNT1), the $S_{EAS}$ was only twice as large (Table 1).

The process of CNT1 and CNT2 doping with N and/or P that includes heat treatment (600–700 °C) leads to an increase in the number of oxygen-containing groups on the surface in the case of CNT1, or to its slight decrease for CNT2. Possibly, a certain contribution into the $S_{EAS}$ value is introduced by defects in the porous structure and hydrophilic–hydrophobic properties of the surface that can change in the course of the doping process.

It was previously shown that severe treatment in nitric acid leads to changes in CNT structure [21]. Shorter fragments of nanotubes are formed, and the specific surface decreases, while the amount of oxygen on the surface increases. As can be seen in Table 1, a significant number of oxygen-containing groups was preserved on the surface of all modified CNT2, and the introduction of doping atoms led only to a slight decrease in oxygen content, possibly due to replacement of oxygen by nitrogen or phosphorus. In addition, acidic treatment leads to significant structure degradation and to the formation of defects at the ends of nanotubes, which can probably contribute to the $S_{EAS}$ value. This was observed for all CNT2.

To determine the activity in ORRs and the effects of the nature of functional groups and oxygen content on the CNT surface on their activity, polarization curves on rotating disk electrodes (RDE) were recorded (Figure 3). RDE measurements were carried out in 0.1 M KOH because ORR catalysis on doped CNT is most pronounced in an alkaline medium, and the shapes of polarization curves allow us to judge the reaction path. The catalytic effect in ORRs can be estimated on the basis of the half-wave potential and the current density at the polarization of 50 mV from the steady-state potential in an $O_2$-saturated electrolyte. It can be seen that functionalization with oxygen-containing groups caused an $E_{1/2}$ shift in the positive direction and an increase of $i_{kin}$ relative to the initial CNT (Table 1, no. 1 for the initial CNT, no. 2 for CNT1, and no. 6 for CNT2). Here, the more oxygen there was on the surface, the larger the $S_{EAS}$ value, but the values of $E_{1/2}$ and $i_{kin}$ increased slightly. Subsequent introduction of P atoms into the structure of CNTs provided a sharp increase in $i_{kin}$, five-fold for CNT1 (Table 1, no. 3) and two-fold for CNT2 (Table 1, no. 7), and also a positive shift in $E_{1/2}$. The value of $i_{kin}$ indicates that doping with nitrogen caused an even greater increase in activity (Table 1, nos. 4, 5 for CNT1 and nos. 8, 9 for CNT2). Thus, CNT-N and dual-doped CNT-NP had the highest activity, irrespective of the method of pre-functionalization, and the $E_{1/2}$ value was 0.79 V versus reversible hydrogen electrode (RHE). This is an increase of 150 mV compared to the initial CNT, and the $i_{kin}$ value exceeded that for the initial CNT (Table 1). After co-doping with N and P atoms, no synergistic effect on activity increase was observed. The limiting diffusion currents for CNT1-N and CNT2-N were higher than the theoretically calculated value for the two-electron ORR (2.8 mA/cm$^2$) *** in an alkaline electrolyte. This indicates the contribution of a four-electron oxygen reduction reaction on the studied materials. The diffusion nature of the observed limiting current showed a linear dependence, as shown in Figure 4.

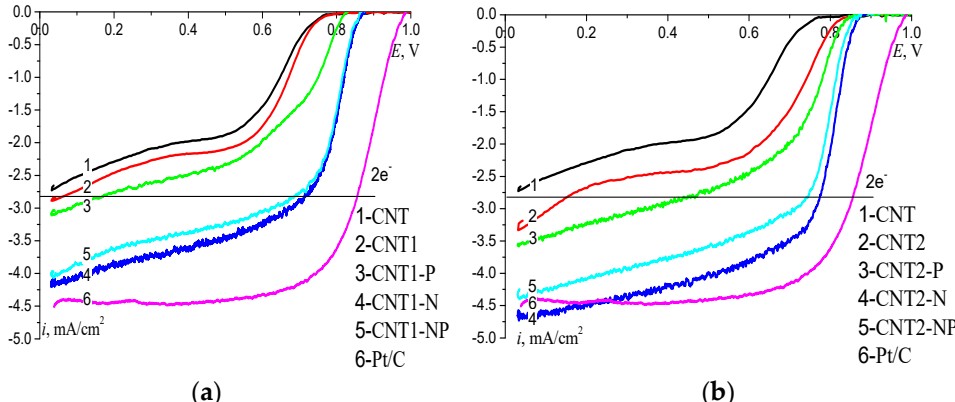

**Figure 3.** Polarization curves of $O_2$ reduction on modified CNTs functionalized: (**a**) in NaOH and (**b**) in HNO$_3$; $O_2$-saturated 0.1 M KOH, the potential scan rate is 5 mV/s, $w$ = 157 rad/s; $m_{cnt}$ = 0.15 mg/cm$^2$.

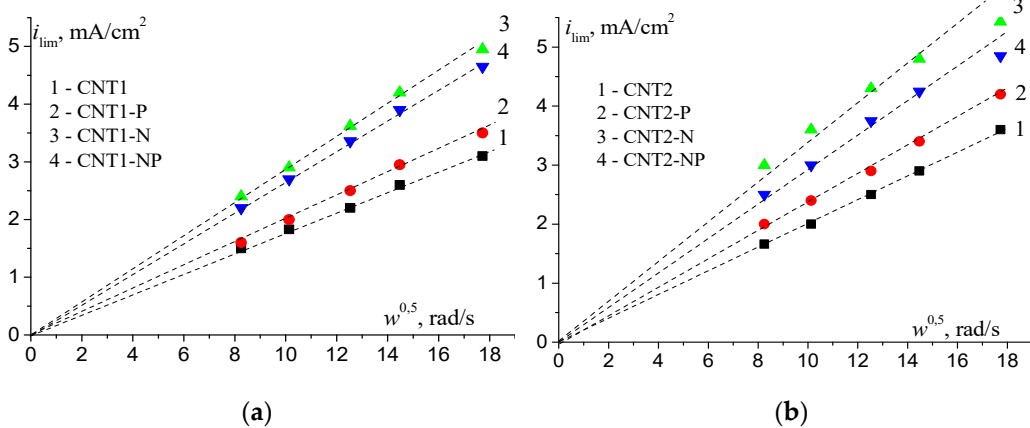

**Figure 4.** Dependence of $i_{lim}$ on $w^{0.5}$ for modified CNTs functionalized: (**a**) in NaOH and (**b**) in HNO₃, O₂-saturated 0.1 M KOH, the potential scan rate is 5 mV/s; $m_{cnt}$ = 0.15 mg/cm². Each dependence is built on the basis of polarization curves at five electrode rotation speeds (w, (rad/s) = 68, 103, 157, 209, 314).

*** The values of the diffusion limiting current density are calculated according to the Koutecky–Levich equation [19].

## 2.3. Accelerated Corrosion Testing

The corrosion stability of modified CNTs was determined by potential cycling in the range of 0.6–1.3 V (versus RHE). The cycling was carried out in an Ar atmosphere. Figure 5 shows that all of the studied nanotubes were stable under experimental conditions. The value of the electrochemically active surface area changed slightly in 1000 cycles. However, the maximum electrochemical surface area decrease of 13 % was observed on CNT2s that had the highest $S_{EAS}$. This is because functionalization in concentrated HNO₃ may cause a change in the porous structure and formation of defects that reduce the stability of CNT during electrochemical corrosion testing. In turn, upon functionalization in NaOH (Figure 5, curve 2), the structure of CNTs remained unchanged (Table 1). Therefore, CNT1s are characterized by higher stability. However, the values of $S_{EAS}$ and $E_{1/2}$ of nanotubes based on CNT1 were much lower than of those based on CNT2 (Figure 5, curve 7). Figure 5b shows the change in half-wave potential during corrosion testing. The smallest change in the half-wave potential was observed for CNT2 doped by N and N+P.

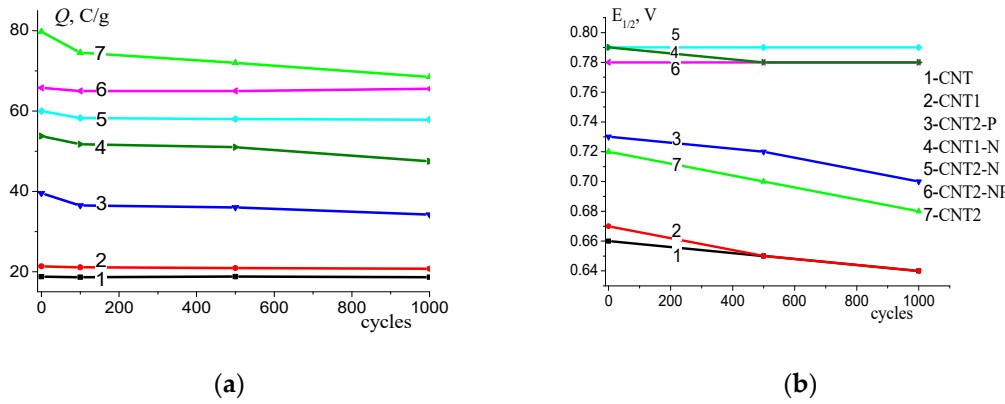

**Figure 5.** (**a**) Variation in electrochemically active surface area; (**b**) variation in half-wave potential during corrosion testing in O₂-saturated 0.1 M KOH. The potential scan rate is 100 mV/s, $m_{cnt}$ = 0.15 mg/cm²; ΔE = 0.6 ÷ 1.3 V (vs. RHE).

Thus, it can be concluded that high activity in ORRs requires the large surface area value provided by functionalization and also the presence of nitrogen and phosphorus to enhance activity and stability. A large electrochemically active surface area observed after CNT functionalization in $HNO_3$ provides an increase in ORR currents only due to an increase in surface area (the current density value in the kinetic range of potentials).

In addition, $E_{1/2}$ was practically unchanged at an increase in $S_{EAS}$. However, the conclusion as to the effect of oxygen-containing groups on ORR acceleration requires additional study.

## 3. Materials and Methods

### 3.1. Chemicals and Materials

Multiwalled CNTs were supplied from Nanotechcenter LLC (Tambov) (>99.0% wt., $S_{BET}$ > 270 m$^2$/g). Initial carbon nanotubes before treatment are denoted as CNTs. Melamine ($C_3H_6N_6$, >99.0%) and TPP (>99.0%) were purchased from Alfa Aesar (Ward Hill, USA) and Merck (Darmstadt, Germany), respectively. Solutions of NaOH (>97.0%) were purchased from Alfa Aesar, and concentrated $HNO_3$ (70% wt.) was purchased from LLC Chimmed (Moscow, Russia). For comparison, commercial (HiSPEC) catalyst 20 wt. % Pt/C was used.

### 3.2. Electrochemical Methods

Studies were carried out by electrochemical methods. Cyclic voltammograms (CV) were obtained at a 100 mV/s scan rate on a stationary electrode. Measurements on a rotating disk electrode (RDE) were performed at a 5 mV/s scan rate at a rotation speed varying from 650 to 3000 rpm in 0.1 M KOH aqueous solution. Experiments were performed on a three-electrode system with a CNT-modified glassy carbon electrode (GCE) (0.126 cm$^2$) sealed in Teflon as the working electrode, a platinum foil as the counter electrode, and an Hg/HgO electrode as the reference electrode. Potential values are given versus RHE. To prepare the catalyst ink, 2.2 μg of modified CNT was dispersed in 500 μL of deionized water, with 5 μL (~150 μg$_{cnt}$ / cm$^2$) of this suspension spread on the GCE surface using a micro-syringe and allowed to dry in air at room temperature. Electrolytes were saturated with oxygen or inert gas by bubbling $O_2$ or Ar prior to the start of each experiment.

$Q_C$ (C/g) is the amount of electricity required to charge the surface of the studied material, and it was determined from the cathode region of the CVs using the following formula:

$$Q_C = \frac{\int i dE}{v} \tag{1}$$

where $v$ is the potential scan rate (V/s), and parameter $Q_C$ is a characteristic of $S_{EAS}$. In addition, CVs without depolarizer (oxygen) in the solution characterized the composition of the active layer on the electrode surface, as manifested by the presence of characteristic peaks.

To determine the activity of the studied materials in ORRs, polarization curves were recorded in $O_2$-saturated electrolytes. Catalytic activity was determined on the basis of half-wave potential ($E_{1/2}$, V), values of the limiting diffusion current density ($i_{lim}$, A/cm$^2$), and current density in the kinetic region, near the steady-state potential ($i_{kin}$, A/cm$^2$).

To determine the corrosion resistance of modified CNTs, the method of accelerated corrosion testing was used. During tests, the potential was cycled in the range 0.6–1.3 V in 0.1 M KOH at a potential rate of 100 mV/s for 1000 cycles. After 100, 500, and 1000 cycles, changes in $Q$ and activity in ORRs were observed.

### 3.3. CNT Modification Methods

Functionalization was carried out using two methods. The first was to treat CNTs in 1 M NaOH at 100 °C for 1 h. These CNTs are denoted as CNT1. The second method was carried out in

concentrated $HNO_3$ at 120 °C for 1 h, in which strong surface oxidation occurred and several types of oxygen-containing groups were formed. In this case, the CNTs are denoted as CNT2.

Nitrogen doping. Functionalized CNT1 or CNT2 was mixed with melamine in a 1:0.7 ratio and milled in a ball mill (Fritsch Pulverisette 7) for 1 h at 800 rpm. The resulting mixture was heated at 600 °C inside a quartz tube furnace for 1 h in an Ar atmosphere. N-doped CNTs are denoted as CNT (1 or 2)–N.

Phosphorus doping. Functionalized CNT1s or CNT2s were mixed with TPP at a 1:5 ratio, dissolved in ethanol, and sonicated for 30 min to form a suspension. The resulting suspension was dried and then heated at 700 °C inside a quartz tube furnace for 1 h in an Ar atmosphere. P-doped CNTs are denoted as CNT (1 or 2)–P.

Dual-doping. To introduce two heteroatoms into the structure of CNTs, two doping techniques were combined. First, CNTs were doped with nitrogen, then the obtained CNT-N were used as the initial material for doping with phosphorus. The obtained CNTs are denoted as CNT (1 or 2)–NP.

### 3.4. Structural Studies

Brunauer–Emmett–Teller (BET) method. BET surface area ($S_{BET}$) and porosity values of the studied materials were determined using the physical sorption method with a Micrometrics ASAP 2020 setup. The adsorption isotherms were measured at 77 K volumetrically examined by nitrogen gas adsorption.

X-ray photoelectron spectra (XPS). XPS were acquired on an Auger spectrometer (Vacuum Generators, UK) with the CLAM2 attachment for measuring XPS spectra. The vacuum in the analyzer chamber was better than 10–8 Torr. An Al anode served as the source of monochromatic radiation (200 W). The peak position was standardized based on the position of a carbon C1s peak with energy of 285.0 eV. For quantitative ratios, we used the coefficients of sensitivity shown in the VG1000 program for spectra processing. The surface layer composition was determined to a depth of 10 nm.

### 4. Conclusions

Pre-functionalization of CNTs with acid or alkali provides an increase in $S_{EAS}$ due to the formation of oxygen-containing groups and active centers on the CNT surface. It can be noted that after $HNO_3$ functionalization, a larger number of oxygen-containing groups of various types formed that provide a large contribution to an increase in $S_{EAS}$ and catalytic activity than the hydroxyl groups formed after NaOH functionalization.

CNTs modified by heteroatoms have high catalytic activity and improved corrosion stability compared to $HNO_3$-functionalized CNTs.

N- and P-doped CNTs can be used as cathode catalysts in alkaline fuel cells, for example in direct alcohol-oxygen ones. The activity of doped CNTs in ORRs in alkaline electrolytes approached that of platinum catalysts. The decrease in overvoltage (half-wave potential) of the ORR reached 150 mV in comparison with the initial CNT.

Due to the high catalytic activity, stability and presence of a large number of active centers on their surfaces, modified CNTs can be used as a substrate for the synthesis of mono and bimetallic catalysts.

**Author Contributions:** Conceptualization, V.B.; Data curation, V.A.; Investigation, I.V. and M.R.; Methodology, O.K.; Resources, V.N. All authors have read and agreed to the published version of the manuscript.

**Funding:** This research received no external funding.

**Acknowledgments:** This work was carried out with the financial support of the RFBR project BRICS_T No.19-53-80033.

**Conflicts of Interest:** The authors declare no conflict of interest.

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
