# Peer review of "Carbon Nanotube Modified by (O, N, P) Atoms as Effective Catalysts for Electroreduction of Oxygen in Alkaline Media"

_catalysts, doi:10.3390/catal10080892_

Round 1
Reviewer 1 Report
The manuscript entitled "Carbon nanotube modified by (O, N, P) atoms as effective catalysts for electroreduction of oxygen in alkaline media" reports research in which the carbon nanotubes surface are doped with elements such as nitrogen and phosphorus. The aim is to improve the stability and catalytic activity of carbon nanotubes respect to the oxygen reduction reaction (ORR). The work is simple and easy to read. The topic covered is consistent with the purpose of the journal.
However, some small revisions are needed before it can be considered for publication.
Here are the suggestions:
Introduction: It lacks a more in-depth description of the oxygen reduction reaction (ORR) being a key aspect of the article. In this way the introduction would be more complete and immediately understandable even for those who do not work directly in this research sector.
Line 70 = a greater description of the characteristics of the carbon nanotubes used is necessary. The melanin and TPP used where they were purchased, what is their purity?
Line 100 = What is the concentration of HNO3?
Fig. 1: What is figure (a) and (b)
Lines 115-117: A more precise description of the dual-doping process is necessary. For example, what quantities are used?
Line 133: correct [17]
Figure 4: Correct the figure. What are YHT2..etc?
Reviewer 2 Report
The manuscript describes the modification of CNT by N and P doping, which increase their catalytic activity in ORR. To increase the impact of the paper, I suggest authors to add some comparison of their results with the results, obtained by other groups, and with well-known reference system (Pt or Pt-C).
Author Response
"Please see the attachment."
